# Pesticides and Their Impairing Effects on Epithelial Barrier Integrity, Dysbiosis, Disruption of the AhR Signaling Pathway and Development of Immune-Mediated Inflammatory Diseases

**DOI:** 10.3390/ijms232012402

**Published:** 2022-10-17

**Authors:** Carla Lima, Maria Alice Pimentel Falcão, João Gabriel Santos Rosa, Geonildo Rodrigo Disner, Monica Lopes-Ferreira

**Affiliations:** Immunoregulation Unit of the Laboratory of Applied Toxinology (CeTICs/FAPESP), Butantan Institute, São Paulo 05503900, Brazil

**Keywords:** pesticides, epithelial barrier integrity, dysbiosis, AhR, immune-mediated inflammatory diseases

## Abstract

The environmental and occupational risk we confront from agricultural chemicals increases as their presence in natural habitats rises to hazardous levels, building a major part of the exposome. This is of particular concern in low- and middle-income countries, such as Brazil, known as a leading producer of agricultural commodities and consumer of pesticides. As long as public policies continue to encourage the indiscriminate use of pesticides and governments continue to support this strategy instead of endorsing sustainable agricultural alternatives, the environmental burden that damages epithelial barriers will continue to grow. Chronic exposure to environmental contaminants in early life can affect crucial barrier tissue, such as skin epithelium, airways, and intestine, causing increased permeability, leaking, dysbiosis, and inflammation, with serious implications for metabolism and homeostasis. This vicious cycle of exposure to environmental factors and the consequent damage to the epithelial barrier has been associated with an increase in immune-mediated chronic inflammatory diseases. Understanding how the harmful effects of pesticides on the epithelial barrier impact cellular interactions mediated by endogenous sensors that coordinate a successful immune system represents a crucial challenge. In line with the epithelial barrier hypothesis, this narrative review reports the available evidence on the effects of pesticides on epithelial barrier integrity, dysbiosis, AhR signaling, and the consequent development of immune-mediated inflammatory diseases.

## 1. Pesticides Disrupt the Integrity of the Epithelial Barriers

The immune system has several levels of barriers capable of containing the initiation and development of autoimmune processes. However, all these barriers can be overcome where there is a genetic predisposition to the impact of various environmental factors that are harmful to the epithelial barrier. The exposome concept argues that surrounding environmental exposures that occur through life are sufficient to cause substantial long-lasting toxicity that—combined with individual genetic components—influence human health (reviewed by [1]) [2,3].

According to the “epithelial barrier hypothesis,” biological and chemical insults from the exposome disrupt the physical integrity of epithelial barriers, triggering alarming epithelial responses and increasing the permeability of the epithelial barrier. These changes determine how the inflammatory immune response will develop and could lead to chronic inflammation, supporting the rise and exacerbation of nearly 2 billion cases of chronic noninfectious diseases, called immune-mediated inflammatory diseases—IMIDs [4,5,6].

Pesticides, along with detergents, nanoparticles, microplastics, household cleaning products, and air pollution, can certainly be added to the list of epithelial barrier-damaging agents to which people are chronically exposed. An increasing trend in pesticide consumption has been observed worldwide (4.2 million metric tons in 2019), with countries such as China, the US, Brazil, and Argentina the largest consumers [7]. Brazilian commodity producers employ a considerable number of pesticides, estimated at 549,280 tons in 2018 [8]. Data from the Brazilian Ministry of Agriculture show that 2837 pesticides were registered in Brazil between 2016 and 2022, including 193 that contained chemicals banned in the European Union (EU) [9,10,11]. Nearly half of all approved products contain active ingredients indexed on the Pesticide Action Network’s list of highly hazardous pesticides, implying that the excessive and dispersed use of pesticides, along with their inappropriate combination, represents an enormous environmental concern [12,13,14,15]. In addition, the use of genetically modified (GM) crops that provide tolerance to synthetic pesticides [16] contributes to their increased consumption.

Pesticides include a wide range of substances, natural or synthetic, classified based on their chemical nature, type of activity, and target organisms, such as insecticides, fungicides, and herbicides, among others [17]. Pesticides are contaminants of emerging concern once they contaminate and alter the quality of local surface and groundwater [18,19,20,21] and also cause indirect impacts in distant areas through transfer among species, the hydrological cycle, and atmospheric circulation [22,23,24,25,26,27,28,29]. Thus, nontarget organisms, including humans, are directly or indirectly exposed to pesticide formulations and their residues.

Research to date has demonstrated that the toxic effects of pesticides are manifested in different ways, from acute and sublethal effects to severe intoxication by chronic exposure to low doses. Evidence from several studies (reviewed by [30,31]) supports carcinogenicity [32,33,34], neurotoxicity [35,36,37,38,39], endocrine and developmental disruption [40,41,42,43,44,45] and metabolic toxicity [46,47,48] as underlying pathogenic mechanisms of pesticides.

Among the extensive list of active ingredients and their commercial formulations registered by the regulatory agencies worldwide, some of them are noteworthy due to their intense use, such as the herbicides glyphosate (GLY, N-(phosphonomethyl) glycine), 2,4-D (2,4-dichlorophenoxyacetic acid), atrazine (2-chloro-4-ethylamino-6-isopropyl-amino-triazine), and the insecticide chlorpyrifos (O, O-diethyl O-(3,5,6-trichloro-2-pyridinyl) phosphorothioate). Pyrethrins are often used as household insecticides and products to control insects on pets or livestock. Several partially resolved mixtures of isomers of allethrin, an insecticide pyrethrin, have become commercially available, including bioallethrin, which remains a common active ingredient in mosquito coils, emanators, vaporizer mats, and other repellent devices [49].

In addition to indirect contamination by the consumption of residues in food and water, pesticides also reach the bloodstream through the skin and mucous membranes of the eyes and respiratory tract, notably in rural workers during the application process [50]. Another critical source of contamination is domestic or occupational exposure to multiple pesticides, especially during pregnancy, which leads to changes in fetal development and serious complications during childhood [51,52,53]. These developmental stages are more susceptible to environmental changes, which is one reason that early chemical exposure events can predispose to health problems faced later in life.

Criswell et al. [54] reported that contaminated breast milk leads to pronounced immune deficiencies in the newborn, increasing the risk of infections, particularly meningitis and inner ear infections in infants. The American Academy of Pediatrics states that children are at higher risk of developing health effects from pesticide exposure than adults [55]. According to an Environmental Working Group (EWG) report, every day, 1 million US children aged 5 years and under consume pesticides that can compromise brain and nervous system development [56].

Interestingly, for pesticide residues to enter and spread into the bloodstream, reaching specific receptors on cellular targets and impairing their homeostasis the first process before plasma absorption and diffusion is transfer through the damaged epithelium of the protective barriers present in the skin, gastrointestinal system and airways. Thus, we hypothesized that the effects of pesticides can be exerted through early changes in the epithelial barrier.

The epithelial barrier consists of a sheet of epithelial cells classified as pseudo-stratified columnar ciliated (which starts in the nasal cavity to the bronchi), squamous epithelial cells (in the alveolar region, allowing gas exchange), stratified keratinized or flattened epithelia (in the skin), and single columnar layer (e.g., gastric mucosa and intestinal villi). In addition to these primary constituents, there are structural proteins involved in the formation of tight junctions (TJs) and adherent junctions (AJs), gap junctions, desmosomes, and hemi-desmosomes. Furthermore, secreted epithelial products such as mucus, antimicrobial peptides, lipid-rich matrix, and epithelial-associated microbiota are essential components of the epithelial barrier [57,58]. Billions of microbes, including bacteria, viruses, fungi, and archaea constitute the microbiome. Bacteria are their main component, and the diversity of communities is estimated at approximately 10^4^ unique strains of bacteria.

The functional epithelium barrier is regulated by specialized TJs, such as occludin and claudins, which are associated with cytoplasmic zonula occludens (ZO) proteins 1, 2, and 3, that constitute a continuous ring around the apicolateral region. Furthermore, communication with AJs such as E-cadherin regulates the structure of the apical-basolateral membrane areas [59]. Primary epithelial barrier dysfunction is characterized by alterations such as disruption of apical cell–cell junctions caused by decreased expression of E-cadherin, β-catenin, occludin and ZOs in a way that is dependent on ROS production, leading to increased epithelial barrier permeability, aberrant epithelial responses, and dysbiosis.

Dysbiosis mainly manifests as decreased microbial diversity and altered microbial community structure, significant attenuation of tryptophan metabolism, and alterations in lipid-related metabolism, including fatty acid metabolism and biosynthesis. Recently, dysbiosis, epithelial barrier hyperpermeability, systemic dissemination of endotoxins and their relationship with chronic inflammation have been regarded as initial factors accounting for a large number of diseases.

In line with well-reported environmental factors in noncommunicable diseases, we present below recent results that show the effects of pesticides on deregulation of the epithelial barrier. Tirelli et al. [60] demonstrated that chlorpyrifos affects epithelial barrier integrity using Caco-2/TC7 cells as an intestinal in vitro model. At the highest dose of 250 μM, impairment of barrier integrity was evidenced, with a reduction in tight junction molecule expression. Ilboudo et al. [61] revealed synergistic effects for five desert locust control pesticides (deltamethrin—DTM, fenitrothion—FNT, fipronil—FPN, λ-cyhalothrin—LCT, and teflubenzuron—TBZ) on human intestinal Caco-2 cell viability and function after exposure for 10 days to 0.1–100 μM of pesticides. They demonstrated a cytotoxic effect of DTM, FNT, FPN, LCT, and TBZ alone or in combination, impacting cell layer integrity and function, alkaline phosphatase (ALP) activity, antioxidant enzyme activity, lipid peroxidation, Akt activation, and apoptosis in an ROS-dependent manner.

Recently, the effects of vitamin E on oxidative stress induced by an organic phosphorus pesticide (phoxim) in the intestinal tissues of Sprague Dawley (SD) rats were demonstrated by Sun et al. [62]. Phoxim significantly reduced jejunum villus height and decreased the mRNA expression of junction protein genes, including occludin and claudin-4. In addition, phoxim increased interleukin-6 (IL-6) and tumor necrosis factor (TNF)-α in jejunum mucosa. Phoxim also increased the DNA expression of total cecal bacteria and *Escherichia coli*, inhibited the DNA expression of *Lactobacillus*, and destroyed the intestinal barrier. Interestingly, vitamin E treatment reduced the effect of phoxim, alleviating the oxidative stress and decreasing the level of TNF-α. The expression of antioxidative stress genes (SOD and GPx2) and DNA expression levels of *Lactobacillus* were significantly increased.

Shah, Sharma, and Banerjee [63] studied the proinflammatory response of organochlorine pesticides (OCPs), β-hexachlorocyclohexane (β-HCH), dichlorodiphenyldichloroethylene (DDE), and dieldrin on human ovary surface epithelial cells (HOSE). They found a high level of ROS production and DNA damage along with upregulation of proinflammatory cytokines such as TNF-α, IL-1β, IL-6, nuclear factor kappa B (NF-kB), and cyclooxygenase (COX)-2 expression. Although emamectin benzoate is considered highly safe because it has specific targets, Niu et al. [64] found that human lung (16HBE) exposed to this pesticide presented inhibition of the proliferation, cytochrome C release, activation of caspase-3/9 and increased Bax/Bcl-2 ratio, which means cytotoxicity is associated with mitochondrial apoptosis. Besides, the DNA damage caused by the emamectin suggests it has a potential genotoxic effect on human lung cells.

Chloropicrin (CP), a soil pesticide, is a strong irritating and lacrimating compound. To elucidate the mechanism of its ocular toxicity, Goswami et al. [65] exposed primary human corneal epithelial (HCE) cells to micromolar doses of CP. CP exposure reduced viability and increased apoptotic cell death with an upregulation of cleaved caspase-3 and poly ADP ribose polymerase. The CP-induced apoptotic cell death pattern included increased expression of heme oxygenase-1 and phosphorylation of histone H2A.X and p53. Furthermore, 4-hydroxynonenal adduct formation is suggestive of oxidative stress, DNA damage, lipid peroxidation, and protein carbonylation.

The oxidative and DNA damage potential of clothianidin (CHN), a member of the neonicotinoid group of insecticides, was investigated in human bronchial epithelial cells (BEAS-2B) by Atlı Şekeroğlu et al. [66]. Their results indicated decreased cell viability in a concentration-dependent manner and DNA single- and double-strand breaks (SSB and DSB), demonstrated by the increase in phosphorylated H2A.X protein foci and p53-binding protein 1 foci. CHN also induced oxidative stress by decreasing reduced glutathione and increasing lipid peroxidation. Zhao et al. [67] investigated the impact of the neonicotinoid pesticide imidacloprid (IMI), detected in the environment and in foods, which is absorbed and metabolized by the intestine. They found that IMI exposure significantly increased intestinal permeability of male Wistar rats exposed to an oral dose of 0.06 mg/kg body weight/day for 90 days, followed by elevated serum levels of endotoxin and inflammatory biomarkers (TNF-α, IL-1β) without any variation in body weight. The disruption of the permeability of the intestinal epithelial barrier was demonstrated by in vitro assays using IMI-treated Caco-2 cells and in silico analyses. Decreased transepithelial electrical resistance, lower expression of tight junction proteins, and disturbance of the PXR-NF-κB p65-MLCK signaling pathway were observed.

Del Castilo et al. [68] found that long-term exposure of mice from pregnancy to adulthood to low doses of a commercial formulation containing GLY induced changes in intestinal barrier integrity, as demonstrated by the altered expression of ZO-1 and ZO-2 and a change in the distribution of syndecan-1 proteoglycan. GLY also led to changes in gut microbiome composition, with consequent change in the intestinal barrier. These data corroborate the findings of Brewster, Warren and Hopkins [69] that showed glyphosate crosses the intestinal–epithelial barrier in SD rats.

As discussed above, the ubiquitous and diverse presence of pesticides in our diet and the persistent environmental exposure that leads to increased insult to the epithelial barrier represents the first step towards systemic inflammation and an important risk factor for developing chronic disease (Figure 1).

## 2. Pesticide-Induced Dysbiosis May Be Associated with AhR Signaling

Throughout the evolutionary process, biological systems have been highly colonized by microorganisms that work in a commensal relationship to maintain the balance in the epithelial barriers. To promote these interactions, microbes play beneficial features due to the release of derived metabolites that interact with membrane cell receptors. Recent studies demonstrate that epithelial-associated microbial communities are involved in maintaining a healthy microenvironment, limiting the colonization of pathogens, playing a key role in the physiology of the epithelium, and modulating the host’s immune response to infections (reviewed by [70,71]).

Fundamentally, intestinal epithelial cells have evolved many strategies for dealing with the dense bacterial loads. Consequently, epithelial cells play a crucial role in limiting bacterial invasion of host tissues, shaping the composition and location of endogenous microbial communities, and coordinating responses of subepithelial immune cell populations. However, in a context of chronic exposure to pesticides, this balance is interrupted and the development of a permeable barrier due to disruption of epithelial integrity leads to microbial dysbiosis. As a consequence of changes in the relative abundance of the intestinal microbiota and in the levels of circulating metabolites after exposure to pesticides, harmful substances pass through the epithelial barrier, causing intestinal inflammation.

The translocation of commensal bacteria and opportunistic pathogens to intraepithelial and subepithelial areas trigger strong innate immune responses through pattern-recognition receptors (PRRs) by pathogen-associated molecular patterns (PAMPs) and damage-associated molecular patterns (DAMPs) on epithelial and immune cells such as keratinocytes, macrophages, innate lymphoid cells (ILCs), and mast cells on the basolateral side of the lamina reticularis. Increased serum concentration of endotoxins, lipopolysaccharide-binding protein (LBP) and soluble CD14 drives local and systemic proinflammatory responses [72].

The epithelium of the skin, gut, and airway barriers express a wide range of receptors located selectively in the membrane of the epithelium cells (e.g., apical region of the endosomal surface or basolateral) that allow them to act as dynamic sensors of the microbial environment (commensal or pathogenic) and as active participants in directing mucosal immune cell responses. The families of PRRs are composed of membrane-bound receptors, such as Toll-like receptors (TLRs) and C-type lectin receptors (CLRs), in both the extracellular and endosomal compartments, cytosolic PRRs that detect exogenous nucleic acids such as retinoic acid-inducible gene I (RIG-I), melanoma differentiation-associated protein 5 (MDA5), interferon inducible gene 16 (IFI16), and absent in melanoma 2 (AIM2). This category would also include the cyclic GMP-AMP synthase (cGAS) stimulator of interferon genes (STING) second messenger system. Finally, the NOD-like receptors (NLRs) are a fourth type of PRRs [73].

Certain NLR proteins, such as NLRP3, NLRP1, and NLRC4, are functionally related in their ability to form inflammasomes together with AIM2, IFI16, and pyrin. Inflammasomes are multiprotein complexes consisting of an NLR and non-NLR, the adaptor protein apoptosis-associated speck-like protein containing a CARD domain (ASC), and the effector protein pro-caspase-1. Once activated, caspase-1 cleaves both pro-IL-1β and pro-IL-18 into their mature and secreted forms, essential actors of the inflammatory and immune responses [74].

Moloudizargari et al. [75] reviewed several studies that demonstrated excessive activation of the NLRP3 inflammasome by pesticides such as paraquat, rotenone, and chlorpyrifos, and the evidence found highlights the correlation between endoplasmic reticulum stress and NLRP3 inflammasome activation with the pathological basis of various inflammatory chronic conditions. Furthermore, it has been demonstrated by [76] that atrazine, widely used in the agricultural sector and toxic to humans, induces oxidative stress and the enhanced expression of Nfkb and IRF1 in spleen, promoting ox-mtDNA formation, which, in turn, stimulates NLRP3 inflammasome activation and pyroptosis.

In addition to having a long-recognized property as a physical and chemical barrier regulating the passage of components between distinct compartments and separating different tissues from the environment, epithelial cells also play important roles in the transition from innate immunity to adaptive immunity. These biological barriers recognize perturbations in their microenvironment, sending signals and secreting products and molecules that profoundly influence the fate of mucosal-resident intraepithelial lymphocytes, regulating immunity and inflammation [77].

Inflammatory memory is not a property unique to immune cells and is also found in skin, lung, and intestine epithelial stem cells. Indeed, it is increasingly recognized as an epigenetic memory for epithelial cells. Skin epithelial stem cells retain chromatin accessibility at key immune response loci after acute inflammation, allowing them to induce an enhanced transcriptional response to a secondary challenge [78].

Once chromatin has been opened, preexisting inflammation-independent transcription factors gain access and can bind to and be retained at memory loci. Together with the associated histone modifications, these preexisting transcription factors preserve accessibility long after inflammation and in the absence of stress-responsive transcription factors reviewed by [79]. More recently, the role of aberrations in the cross talk of epithelial immune cells in cancer susceptibility has been demonstrated. Although the memory inflammation mitigates subsequent tissue damage, it also potently predisposes tumor-generating cells to malignant transformation upon an oncogenic assault [80].

These recent data lead us to consider that one of the deleterious consequences of persistent exposure to environmental pesticides is the accumulation of epigenetic marks in inflammatory loci and the subsequent predisposition to chronic inflammatory diseases and cancer.

The gastrointestinal (GI) tract harbors up to 70% of the body’s lymphocyte population and is one of the most important interfaces of contact with exogenous molecules. This community is composed of natural intestinal intraepithelial lymphocytes (IELs), and CD8αα+CD4+ IELs and TCRαβ-bearing CD8αβ+ induced IELs, serving as long-lived effector memory cells to confer protection against gut-associated pathogens and intestinal tolerance [81]. Further, submucosal innate lymphoid cells (ILCs) are essential for the integrity and homeostasis of the lung mucosal barrier [82]. Also, Moreau et al. [83] discovered that TGF-β-activated αvβ8 regulatory T cells residing in skin possessed a tissue-specific transcriptional profile that potentiates keratinocyte cell–cell communication that influences epithelial cell biology.

Interestingly, the sensory nervous system is highly concentrated in epithelial barriers and broadly responsive to many stimuli [84,85]. As an example, the enteric nervous system in the gut, which is composed of three neuron types (intrinsic primary afferent neurons—IPANs, vasoactive intestinal peptide neurons—VIPs, and cholinergic neurons) influences the level of inflammation through enteric neurons and extrinsic neural connections, particularly vagal and sympathetic innervation of the GI tract. The enteric nervous system coordinates with the intestinal epithelium and constitutes an anatomical and functional unit called the neuronal–glial–epithelial unit (reviewed by [86]).

Recently, cross talk between the neuronal–glial–epithelial unit and the microbiota has been shown to modulate epithelial barrier permeability, intestinal development, and the immune response. Within the gut, the microbiota can produce neuroactive compounds, such as neurotransmitters (for example, γ-aminobutyric acid—GABA, noradrenaline, dopamine, and 5-hydroxytryptamine—5-HT, called serotonin), amino acids and amino acid metabolites (e.g., L-tryptophan (L-Trp), a precursor of metabolites such as L-kynurenine (L-Kyn), serotonin, indolic, and tryptamine) and microbial metabolites as short-chain fatty acids (SCFAs) and 4-ethylphenylsulfate.

The highly bidirectional communication between the brain and the gut is markedly influenced by the microbiome through integrated immunological, neuroendocrine and neurological processes [87] that ultimately affect the nervous system and may directly influence neuropsychiatric disorders associated with development (e.g., autism spectrum disorder (ASD) and schizophrenia), mood (e.g., depression and anxiety) and neurodegeneration (e.g., Parkinson’s disease (PD), Alzheimer’s disease (AD), and multiple sclerosis (MS) (reviewed by [88]) [89].

In the airway wall, a dynamic epithelial–immune–neuronal unit plays a critical role in maintaining mucosal immune homeostasis as well as facilitating host defense against inhaled pollutants such as pesticides [90]. Atopic dermatitis (AD) is a common inflammatory dermatosis that affects up to 30% of children and 10% of adults worldwide. The primary defect of AD is mainly a defect in the epidermal barrier that triggers immune deregulation in the skin. According to recent research, such phenomena are closely related to microbial skin dysbiosis [91].

The environmental sensor and transcription factor aryl hydrocarbon receptor (AhR) is a cytosolic ligand-dependent transcription factor that belongs to the superfamily of basic helix-loop-helix (bHLH), with the PER-ARNT-SIM (PAS) domain acting as a detector of endogenous and exogenous factors. AhR activation leads to CYP1A expression, which encodes phase I enzymes responsible for detoxification. Beside its roles in response to toxic pollutants, recently, AhR has been mainly studied due to its involvement in physiological and immune processes [92].

AhR also plays a biological role in immune modulation and differentiation, especially in B cell expansion, maturation, and differentiation. The AhR-dependent activation of B1a CD5+ B cells generates a decrease in natural IgM production [93,94]. These cells in the peritoneal intestine and pleura are essential for antibacterial response in infancy and old age when the immune system is developing or declining. It was recently described by Diny et al. [95] that AhR controlled longevity, tissue adaptation and functions of eosinophils, and gene expression in the small intestine.

Notably, in addition to expression in epithelial-associated leukocytes [96], AhR is expressed primarily in the epithelial barrier of the lungs, skin, and intestine [97,98,99] where it protects from inflammatory damage by maintaining intestinal stem cell (ISC) homeostasis. Metidji et al. [100] found that the AhR activation in intestinal epithelial cells (IECs) by dietary ligands restored barrier homeostasis, protected the stem cell niche, and prevented tumorigenesis via transcriptional regulation of *Rnf43* and *Znrf3*, E3 ubiquitin ligases that inhibit Wnt-β-catenin signaling and restrict ISC proliferation.

Combining a series of human studies, in vivo mouse models, and in vitro analyses, Postal et al. [101] demonstrated the protective effect of AhR activation in the intestine, targeting particularly tight junctions and cytokine expression. They found that pretreatment of Caco-2/TC7 cells with the AhR agonist βNF prevented chemically induced damage to the intestinal epithelium monolayer by a mechanism involving an increase in the levels of occludin and tricellulin dependent on increased expression of AhR target genes CYP1A1 and CYP1A2.

Corroborating its protective features, Yu et al. [102] found that AhR activation by 6-formylindolo(3,2-b)carbazole (FICZ) ameliorates the dextran sulfate sodium (DSS)-induced disruption of intestinal barrier function by decreasing paracellular permeability and maintaining TJ barrier integrity. The lack of AhR signaling in psoriasiform inflammation in mice exacerbates the severity of inflammatory response by unleashing excessive production of chemokines and cytokines in response to proinflammatory stimuli [103]; while CYP1A1 enzymatic activity is a critical regulator of beneficial AhR signaling in the context of inflammation upregulating the expression of barrier-related proteins and accelerating terminal keratinocyte differentiation [104].

AhR activation leads to maintenance of epithelial barrier integrity, preventing permeability dependent on junctional changes, but in dysbiosis, even under an astonishing quantity of AhR ligands, metabolites such as the kynurenic acid (KynA) have a protective effect on the intestinal epithelium via activating AhR [105]. Taddese et al. [106] found that gut bacteria have the potential to modulate the expression of biotransformation pathways in colonic epithelial cells in an AhR-dependent manner.

The impact of pesticides on the digestive tract is of particular interest, leading to alterations in intestinal microbiota communities and dysbiosis [107] once it is the first organ in contact with ingested contaminants and is responsible for the development and function of the nervous system. Recently, an extensive analysis of the effects of pesticides on the microbiota of different organisms was carried out by Giambò, Teodoro, Costa, and Fenga [108]. Data from a total of 117 articles on the action of pesticides were reviewed (particularly glyphosate, triazine, 2,4-D, chlorpyrifos, organochlorines, insecticides, fungicides, and a mixture of them) in the gut microbiota of various species, such as honeybees, wasps, rats, mice, mussels, crabs, oysters, marine turtles, zebrafish, carp, birds of the family Phasianidae, *Bombyx mori*, silkworms, earthworms, *Drosophila melanogaster*, mosquitoes and German cockroaches, among others.

From the quantitative and qualitative data of these reviewed articles, a common pathogenic mechanism was proposed, highlighting that pesticides disrupt the typical composition and functionality of the gut microbiome with modulation of different phylum types. These abnormalities lead to significant metabolic imbalances, especially in energy- and fat-related metabolic pathways, oxidative stress, and DNA damage. The impaired microbiota after acute exposure to pesticides alters the development of intestinal structures, including decreased cell proliferation in the crypts, the expression of stem cell markers as well as secretory cell markers, reducing the number of goblet cells and protein mucin and decreasing expression levels of tight junction markers [108].

Numerous in vitro and in vivo studies have shown that long-term exposure to pesticides may disturb the microbiota, leading to dysbiosis in both humans and animals. AhR ligands improved intestinal barrier affecting the integrity of the apical junctional complex. Therefore, the dysbiosis impairs epithelial barrier function and the outcome of the AhR signaling. The AhR pathway reflects a prototypic pathway at the interface microbiota–epithelial barrier–metabolism and immune functions. Finally, we can extend in this model which role AhR might play in a putative pesticide–epithelial barrier–dysbiosis axis (Figure 2).

## 3. Immune-Mediated Inflammatory Disease Development Is Enhanced by Pesticides

In addition to the local targeting of tissue-specific AhR via microbial metabolites, the systemic entry of metabolites formed from the barrier breached by pesticides results in far-reaching implications in other organs such as the brain, lung, and skin [109,110,111]. These abnormal phenomena represent an early and critical event for chronic immune responses to environmental onsets in the epithelia. Efforts would be desirable to expand investigations into the mechanisms of how exposures translate into making individuals, especially pregnant women, children and the elderly, more susceptible to chronic inflammatory disease development.

The IMIDs comprise a clinically diverse group of chronic conditions for which there is no cure and are often accompanied by multiple comorbidities. IMIDs include rheumatoid arthritis (RA), inflammatory skin conditions (including psoriasis and atopic dermatitis), inflammatory bowel disease (IBD, including Crohn’s disease and ulcerative colitis), airway inflammation (bronchitis, asthma, or rhinitis), and metabolic syndrome (including obesity, type 2 diabetes and insulin resistance, steatosis, and nonalcoholic fatty liver disease).

Scientific advances with the use of animal models that mimic several diseases and the application of multi-omic techniques like (epi)genomic, transcriptomic, proteomic, and metabolomic methods have changed the concept of the pathophysiology of IMIDs, as they have revealed the crucial role of environmental factors as pesticides alone or in combination with other pollutants, in their triggering, progression, and worsening, in addition to the classic genetic and immunological factors (Figure 3).

Asthma or similar respiratory pathologies are recognized as inhomogeneous diseases since multiple subtypes and different factors have been linked to increased susceptibility to these conditions. They have been increasingly prevalent in countries where the use of multiple chemicals for crop protection and home disinfestation has increased. To illustrate the association between pesticide exposure and symptoms of airway inflammation, bronchial hyperreactivity, and asthma, in the AGRICAN cohort, a large French agricultural population, self-reported doctor-diagnosed asthma was analyzed by Baldi et al. [112]. They described an increased risk of allergic asthma associated with exposure to pesticide (such as fungicides) exposure in the early years of life and especially on farms producing wine, sugar beet, fruit and vegetables. In contrast, no significant excess in risk was observed for nonallergic asthma or in livestock farmers.

Moreover, Henneberger et al. [113] identified positive exacerbation–pesticide associations of symptoms for the herbicide pendimethalin and the insecticide aldicarb. Among the adult pesticide applicators with active asthma, 22% exhibited exacerbation of symptoms. In a cross-sectional study covering female farm workers in Africa, the prevalence of ocular–nasal symptoms was positively associated with pesticide-sprayed fields [114]. Patel et al. [115] found in 11,210 US primary farm operators positive associations between the use of pesticides, notably the use of 2,4-D, and lifetime allergic rhinitis and current asthma.

Among the various risk factors for the development of RA, a systematic review of relevant literature between 1956 and 2021 and a meta-analysis were conducted to investigate the association between pesticide exposure and RA [116]. A total of eight studies were eligible for inclusion (two cohort studies, four case–control studies, and two cross-sectional studies) and showed that exposure to insecticides (especially fonofos, carbaryl, and guanidines) contributes to an increased risk of RA.

Obesity is one of the most severe global public health issues, especially because it increases the risk of many chronic diseases, such as cardiovascular diseases, diabetes, neurological disorders, and even some types of cancer. In 2016, the World Health Organization estimated that 39% of adults were overweight worldwide, of whom 13% were obese. The control and management of obesity require identifying the potential factors associated with it, and environmental chemical exposure might be one of them.

Pesticides can regulate the weight [117,118,119] by modifying metabolism and homeostatic set points, affecting appetite regulation, and altering lipid metabolism [47,120], which stimulates adipocyte hypertrophy and promotes adipogenic pathways that target hyperplasia of fat cells [121,122] and steatosis [123], predisposing, initiating or exacerbating weight gain.

Type 2 diabetes mellitus (T2D) is clinically defined by elevated blood glucose levels caused by several disruptions in the molecular physiology that regulates glucose homeostasis. Emerging scientific evidence indicates that T2D might be affected by environmental pollutants, since its prevalence accounts for 90–95% of diabetes mellitus and has more than doubled from 30 years ago [124]. Another factor that supports chemical exposure dependence is the increasing cases in children, young adults, and nonobese individuals, which is not fully explained only by conventional factors such as age, unhealthy diet, obesity, genetic mutation, and heredity. Among the pesticides, the organochlorines (OCPs) and their metabolites are notably suspected to impart a considerable risk of developing T2D and its comorbidities.

Cross-sectional studies in adult populations have established a positive relationship between exposure to pesticides and T2D, insulin resistance and dyslipidemia in several countries, such as Algeria [125], India [126], China [127], South Korea [128], Canada [129], and the US [130,131].

IMIDs that develop in the gastrointestinal (GI) tract share similar pathophysiology with similar manifestations. These disorders, including inflammatory ulcerative colitis (UC), Crohn’s disease (CD), celiac disease, and eosinophilic gastroenteritis, are characterized by chronic gut inflammation of unknown etiology. Factors influencing the occurrence and development include aberrant immune responses, genetic susceptibility, intestinal dysbiosis, persistent intestinal infections, chronic intestinal mucosal barrier injury, poor diet, and unquestionably chronic exposure to pesticides [132].

Studies proving the direct action of pesticides triggering inflammatory diseases of the digestive system were not found. However, an indirect effect of pesticides causing chronic intestinal mucosal barrier injury and dysbiosis that leads to the production of metabolites by the microbiota capable of inducing inflammation has been demonstrated [133]. Interestingly, pesticides in the food chain can impair neuroendocrine functions in the gut by compromising hormone production produced and secreted by the enteroendocrine cells (EECs), destabilizing cross talk between the neuronal–glial–epithelial unit and playing roles ranging from central regulation of appetite and food intake to inflammation and brain disorders such as anxiety and depression [132,134].

Humans are exposed to a multitude of environmental pollutants, including pesticides through the environment and contaminated food. Long-term exposure to these contaminants is associated with triggering, progression and aggravation of IMIDs. Given the intimate association between pesticide-induced epithelial barrier breakdown and microbiota deregulation and the crucial role of dysbiosis in various human diseases, it is both intriguing and challenging to unravel the complex underlying molecular mechanisms and their significance for human health and disease. Microbiota can transform and metabolize xenobiotic compounds and transform them into endogenous AhR ligands. AhR is mainly expressed in barrier organs such as lung, skin and intestine. In a dysbiosis context, the availability of ligand changes, which often alters the constitutive AhR signaling necessary for structural integrity and epithelial barrier function, as well as for immunity. In this review, we have highlighted the hypothesis that the disruption of AhR signaling is a common link between these processes.

## Figures and Tables

**Figure 1 ijms-23-12402-f001:**
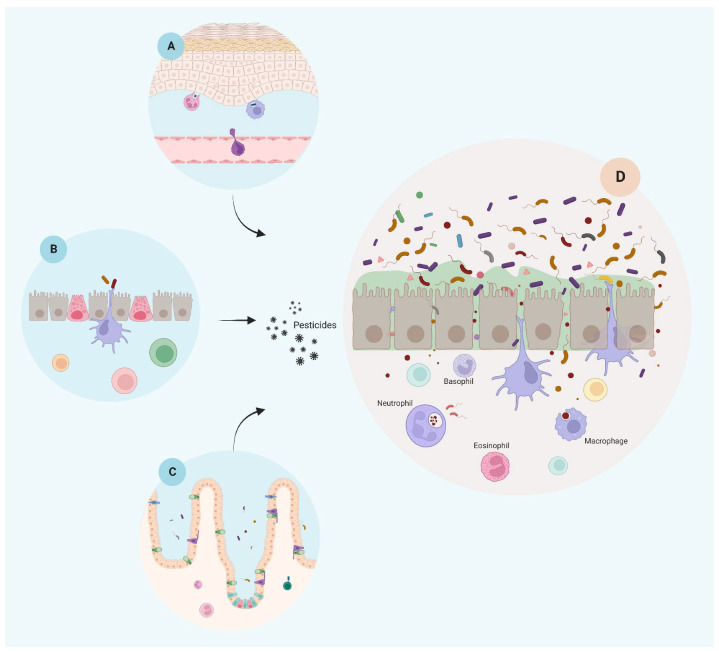
Epithelial barrier integrity is negatively affected by pesticide exposure. On the left, an illustrative representation of the main epithelial barriers in their natural state, i.e., skin (**A**), lung (**B**), and gut (**C**), before exposure to pesticides and potentially other environmental stressors. Following chemical exposure (pesticides), a schematic representation is shown of a generic epithelial tissue presenting several inflammatory response outcomes after disruption of the barrier integrity due to pesticide exposure, such as loss of mucus and villi, increase of barrier permeability by cellular junction rupture, leaking, dysbiosis, and inflammatory response (**D**). Created with BioRender.com.

**Figure 2 ijms-23-12402-f002:**
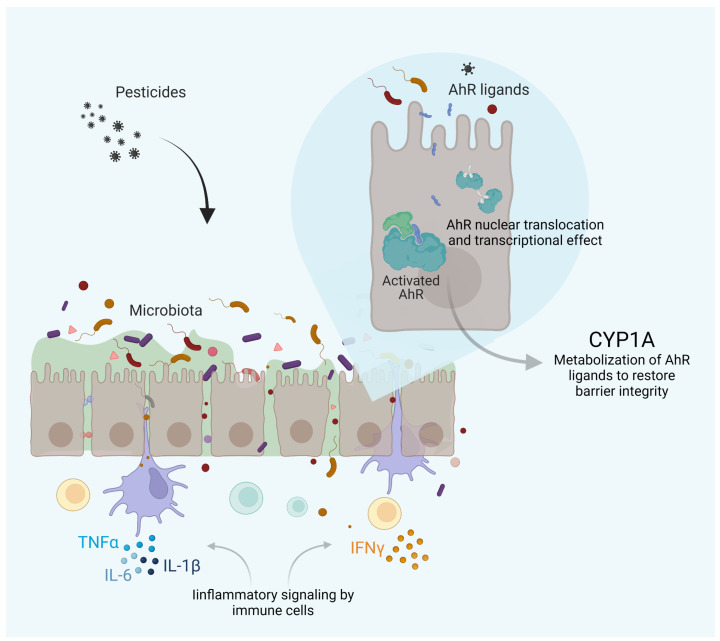
AhR signaling in pesticide-induced dysbiosis, and inflammation. AhR acts as a ligand-activated sensor of endogenous and exogenous factors. After dimerization with AhR nuclear translocator (ARNT), it binds to response elements in the DNA and promotes the transcription of genes involved in physiological pathways, notably CYP1A expression, contributing to maintain the epithelial barrier integrity. However, in dysbiosis, deregulated production of microbiome metabolites is responsible for altered stimulation of the AhR pathway, promoting inflammation. Created with BioRender.com.

**Figure 3 ijms-23-12402-f003:**
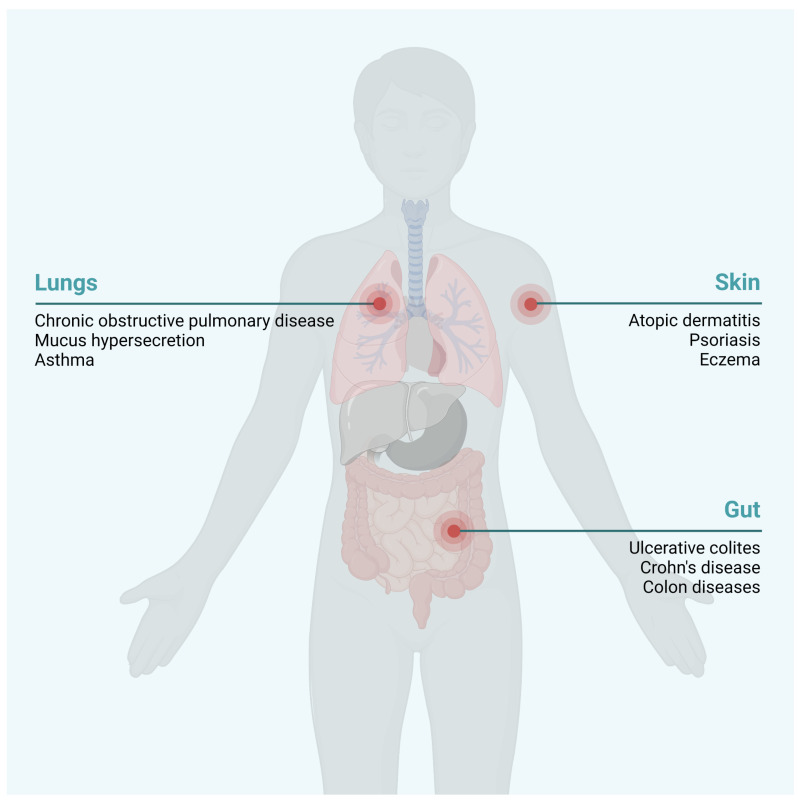
Chronic inflammatory diseases potentially developed by pesticide exposure. Pesticide exposure has been linked to the development of several immune-mediated inflammatory diseases (IMIDs), mainly in the skin, airway system, and gut. Created with BioRender.com.

## Data Availability

Not applicable.

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
