# Peer review of "Pesticides and Their Impairing Effects on Epithelial Barrier Integrity, Dysbiosis, Disruption of the AhR Signaling Pathway and Development of Immune-Mediated Inflammatory Diseases"

_ijms, 2022, doi:10.3390/ijms232012402_

Round 1
Reviewer 1 Report
General comments
Title: …AhR signaling as a possible common link… To my opinion, this connection is not shown by the text at least it is shown in chapter 2 for dysbiosis only. Chapter 1 is a list of pesticides which may cause a breakdown of the epithelial barrier. And chapter 3 describes IMIDs. Only the very last sentence of chapter 4 (and of the whole text) tries to come back to the title.
Thus, I would suggest rewording the title for a more generic message:
Pesticides and their impairing effects on the epithelial barrier integrity, dysbiosis, disruption of the AhR signaling pathway and development of chronic inflammatory diseases
Abbreviations: There are plenty of abbreviations in the text, and some were not explained. So, I would suggest adding a list of all abbreviations i.e. between the text and the references.
Structure of chapter 1: This chapter which is headed by “…. of the epithelial barrier” also lists other health effects of pesticides. For example, glyphosate is first mentioned on page 2, line 87, in the context of brain damage. Then, on page 5, line 221, glyphosate again is cited as dangerous for the intestinal barrier function. Furthermore, data from animal studies, in vitro-studies and findings in man are simply collected without a clear separation. Chapter 1 should urgently be subdivided i.e. according to results from animal studies, in vitro-studies and data definitively found in patients or according to the different classes of pesticides.
Why did you also list ionic liquids? I could not find a hint on their use in industrial preparations of pesticides. If it is so, it does not make sense to also add this to chapter 1.
Special comments
Page 1, line 41: alarming responses
Page 2, line 84: remove and before Atrazine
Page 2, lines 84/85: 2-chloro-4-ethylamino-6-isopropylamino-triazine; s seems to be needless (no chiral structure, no sulfur atom)
Page 2, line 85: set a comma instead of semicolon after …triazine)
Page 2, line 97: classified it as …
Page 3, line 116: set full stop after [58] instead of semicolon and go on with “These functions are required…”
Page 3, lines 132-136: This paragraph has its message doubled, please reword.
Page 3, lines 137-144: This paragraph is too long for one sentence and is not understandable on first sight. Please form more sentences to give more structure and to help the reader understanding this list of all constituents of the epithelial barrier.
Page 3, lines 145-148: Both parts of the paragraph separated by semicolon do not form a unique sentence and must be reworded. At least, a verb has to be added in part two.
Page 4, line 151: remove semicolon after production
Page 4, line 163: pesticides-induced – remove s
Page 4, line 169: Remove 4. (At least, I could not find 1., 2., 3.)
Page 4, line 170: …total fecal …
Page 4, line 184: …cytotoxicity is associated…
Page 4, line 202/203: Set full stop after p53 and go on with: “Furthermore, 4-hydroxynonenal adduct formation is suggestive …”
Figure 1, last line of legend: Could be omitted with a list of abbreviations as mentioned above.
Page 6, line 243: …protective to the immune system…
Page 7, line 292: …amino acids (e.g. tyramine and tryptophan)…; tryptophan indeed is an amino acid, but tyramine is an amine which is formed out of the amino acid tyrosine. Please correct.
Page 7, line 302: For readers who are not familiar enough with it, please explain NOD and RIG (i.e. via a list of abbreviations, see above)
Page 7, line 315: Besides its role… (remove s)
Page 7, line 322-324: This sentence is not to understand, please reword.
Page 7, line 325-332: This sentence is rather long and the last part (…stem cells and regeneration…) is not correctly linked to the first part. Please reword.
Page 7, line 343-344: Set a full stop after reticularis, remove and and go on with “Increased serum…” remove that in line 344 and add s to drive.
Page 8, line 356: There was reviewed data…
Page 8, line 358: set comma instead of semicolon after Chlorpyrifos
Page 8, line 359-361: listing of insects is mixed mode plural and singular; please unify.
Page 8, line 361: set and after mosquitoes,
Page 8, line 376: normally drives…
Page 8, line 383: in promoting… Remove or
Page 10, line 408: … progression, and worsening… of what? Please specify.
Page 10, line 429: RA instead of AR
Page 10, line 441-445: sentence is not clear, may be too long; please reword or make two sentences.
Page 12, line 499: ligands changes, remove s
Page 12, line 501: …as well as for immunity
Page 12, line 507-509: 3 spaces are missing
Ref 1.: A Practical Document; something is wrong with the title of the Journal, please correct
Ref 20: is abbreviation correct?
Remove line between Ref. 28 und 29
Ref 30: is abbreviation correct?
Ref. 50: please complete with article number 4605
Ref. 53: please complete with article number 22
Ref. 56: please complete with article number
Ref. 70, line 712: Chlorpyrifos
Ref. 82: something is wrong with this citation: doi did not work, e338-8 is not plausible – please check.
Ref. 89: complete with article number
Ref. 100: complete with article number
Ref. 101: abbreviation in the text different AHR – AhR, please check
Ref. 105: see Ref. 101
Ref. 116: see Ref. 101
Ref. 118: please complete with article number
Ref. 125: please complete with number of last page
Ref. 132: please complete with article number
Ref. 135: please complete with article number
Ref. 136: please complete with article number
Author Response
International Journal of Molecular Sciences
Section: Molecular Toxicology
Special Issue: Impact of Environmental Contaminants in Diseases of Aquatic Organisms and Human Health
Manuscript ID: ijms-1920425 R1
POINT-BY-POINT RESPONSE TO REVIEWERS
First of all, we appreciate you taking the time out to share your comments with us; we value and respect your opinion, and we agree that improvements must be made to polish up the manuscript. We have made a considerable effort to set the central point very clearly and systematically organized the manuscript text as far as we thought it would improve it.
Responses to Reviewers Round 1
Reviewer #1
General comments
Title: …AhR signaling as a possible common link… To my opinion, this connection is not shown by the text at least it is shown in chapter 2 for dysbiosis only. Chapter 1 is a list of pesticides which may cause a breakdown of the epithelial barrier. And chapter 3 describes IMIDs. Only the very last sentence of chapter 4 (and of the whole text) tries to come back to the title.
Thus, I would suggest rewording the title for a more generic message:
Pesticides and their impairing effects on the epithelial barrier integrity, dysbiosis, disruption of the AhR signaling pathway and development of chronic inflammatory diseases
Abbreviations: There are plenty of abbreviations in the text, and some were not explained. So, I would suggest adding a list of all abbreviations i.e. between the text and the references.
A list of abbreviations was updated.
Structure of chapter 1: This chapter which is headed by “…. of the epithelial barrier” also lists other health effects of pesticides. For example, glyphosate is first mentioned on page 2, line 87, in the context of brain damage. Then, on page 5, line 221, glyphosate again is cited as dangerous for the intestinal barrier function. Furthermore, data from animal studies, in vitro-studies and findings in man are simply collected without a clear separation. Chapter 1 should urgently be subdivided i.e. according to results from animal studies, in vitro-studies and data definitively found in patients or according to the different classes of pesticides.
Why did you also list ionic liquids? I could not find a hint on their use in industrial preparations of pesticides. If it is so, it does not make sense to also add this to chapter 1.
Special comments
Page 1, line 41: alarming responses
Corrected.
Page 2, line 84: remove and before Atrazine
Corrected.
Page 2, lines 84/85: 2-chloro-4-ethylamino-6-isopropylamino-triazine; s seems to be needless (no chiral structure, no sulfur atom)
Corrected.
Page 2, line 85: set a comma instead of semicolon after …triazine)
Corrected.
Page 2, line 97: classified it as …
Corrected.
Page 3, line 116: set full stop after [58] instead of semicolon and go on with “These functions are required…”
Corrected.
Page 3, lines 132-136: This paragraph has its message doubled, please reword.
We edited the manuscript to avoid redundancy.
Page 3, lines 137-144: This paragraph is too long for one sentence and is not understandable on first sight. Please form more sentences to give more structure and to help the reader understanding this list of all constituents of the epithelial barrier.
We made a few changes to make it more clear. Please check the edited manuscript.
Page 3, lines 145-148: Both parts of the paragraph separated by semicolon do not form a unique sentence and must be reworded. At least, a verb has to be added in part two.
We made a few changes. Please check the edited manuscript.
Page 4, line 151: remove semicolon after production
Corrected.
Page 4, line 163: pesticides-induced – remove s
Corrected.
Page 4, line 169: Remove 4. (At least, I could not find 1., 2., 3.)
Claudin-4 is the protein nomenclature.
Page 4, line 170: …total fecal …
In the excerpt the quote meant to refer to the “total cecal bacteria”, as written.
Page 4, line 184: …cytotoxicity is associated…
Corrected.
Page 4, line 202/203: Set full stop after p53 and go on with: “Furthermore, 4-hydroxynonenal adduct formation is suggestive …”
Corrected.
Figure 1, last line of legend: Could be omitted with a list of abbreviations as mentioned above.
A list of abbreviations was added to text.
Page 6, line 243: …protective to the immune system…
Corrected.
Page 7, line 292: …amino acids (e.g. tyramine and tryptophan)…; tryptophan indeed is an amino acid, but tyramine is an amine which is formed out of the amino acid tyrosine. Please correct.
Corrected.
Page 7, line 302: For readers who are not familiar enough with it, please explain NOD and RIG (i.e. via a list of abbreviations, see above)
A list of abbreviations was added to text.
Page 7, line 315: Besides its role… (remove s)
Corrected.
Page 7, line 322-324: This sentence is not to understand, please reword.
We adjusted the sentence. Please check the edit manuscript.
Page 7, line 325-332: This sentence is rather long and the last part (…stem cells and regeneration…) is not correctly linked to the first part. Please reword.
We adjusted the sentence. Please check the edit manuscript.
Page 7, line 343-344: Set a full stop after reticularis, remove and and go on with “Increased serum…” remove that in line 344 and add s to drive.
Corrected.
Page 8, line 356: There was reviewed data…
Corrected.
Page 8, line 358: set comma instead of semicolon after Chlorpyrifos
Corrected.
Page 8, line 359-361: listing of insects is mixed mode plural and singular; please unify.
Corrected.
Page 8, line 361: set and after mosquitoes,
Corrected.
Page 8, line 376: normally drives…
Corrected.
Page 8, line 383: in promoting… Remove or
Corrected.
Page 10, line 408: … progression, and worsening… of what? Please specify.
It reflects the idea of progression and worsening of immune-mediated inflammatory diseases (IMIDs). Now, we specify in the text.
Page 10, line 429: RA instead of AR
All the abbreviations concerning “rheumatoid arthritis” are referred to as “RA”.
Page 10, line 441-445: sentence is not clear, may be too long; please reword or make two sentences.
We adjusted the paragraph. Please check the edit manuscript.
Page 12, line 499: ligands changes, remove s
Corrected.
Page 12, line 501: …as well as for immunity
Corrected.
Page 12, line 507-509: 3 spaces are missing
To be corrected at the final text edition.
Ref 1.: A Practical Document; something is wrong with the title of the Journal, please correct
Corrected.
Ref 20: is abbreviation correct?
Yes, it is correct.
Remove line between Ref. 28 und 29
Corrected.
Ref 30: is abbreviation correct?
Yes, it is correct.
Ref. 50: please complete with article number 4605
Corrected..
Ref. 53: please complete with article number 22
Corrected.
Ref. 56: please complete with article number
Corrected.
Ref. 70, line 712: Chlorpyrifos
Corrected.
Ref. 82: something is wrong with this citation: doi did not work, e338-8 is not plausible – please check.
Corrected.
Ref. 89: complete with article number
Corrected.
Ref. 100: complete with article number
Corrected.
Ref. 101: abbreviation in the text different AHR – AhR, please check
We kept the original article title.
Ref. 105: see Ref. 101
We kept the original article title.
Ref. 116: see Ref. 101
We kept the original article title.
Ref. 118: please complete with article number
Corrected.
Ref. 125: please complete with number of last page
The original reference article does not mention page numbers.
Ref. 132: please complete with article number
Corrected.
Ref. 135: please complete with article number
Corrected.
Ref. 136: please complete with article number
Corrected.

Reviewer 2 Report
Interrupción de la señalización de AhR como posible vínculo común entre pesticidas, ruptura de la integridad de la barrera epitelial, disbiosis y enfermedades inflamatorias crónicas. Es un tema importante a tratar. Sin embargo, considero que algunos aspectos deberían mejorarse antes de su aprobación.
1) La figura 1 debe incluir células caliciformes y células M, que reaccionan inmunológicamente con patógenos y compuestos químicos.
Describa mejor la leyenda de la figura 1, incluyendo el nombre de las abreviaturas.
2) Describir con mayor detalle el mecanismo por el cual CYP1A permite mantener la integridad celular.
CYP1A, encontrado en mayúsculas y minúsculas, correcto.
Figura 2, la imagen es confusa, no se indica claramente la secreción de las células, el estímulo que reciben y que citoquinas les son propias. Es necesario utilizar flechas.
3) Es fundamental en un proceso inflamatorio explicar el efecto de los pesticidas u otras sustancias sobre la expresión del complejo inflamasoma, debe incluirse en la revisión.
4) Considero necesario hacer una descripción por familias de plaguicidas como los piretroides en cada ítem.
5) es importante mencionar en la revisión el efecto de la disbiosis en el sistema nervioso.
Author Response
International Journal of Molecular Sciences
Section: Molecular Toxicology
Special Issue: Impact of Environmental Contaminants in Diseases of Aquatic Organisms and Human Health
Manuscript ID: ijms-1920425 R1
POINT-BY-POINT RESPONSE TO REVIEWERS
First of all, we appreciate you taking the time out to share your comments with us; we value and respect your opinion, and we agree that improvements must be made to polish up the manuscript. We have made a considerable effort to set the central point very clearly and systematically organized the manuscript text as far as we thought it would improve it.
Responses to Reviewers Round 1
Reviewer #2
Interrupción de la señalización de AhR como posible vínculo común entre pesticidas, ruptura de la integridad de la barrera epitelial, disbiosis y enfermedades inflamatorias crónicas. Es un tema importante a tratar. Sin embargo, considero que algunos aspectos deberían mejorarse antes de su aprobación.
1) La figura 1 debe incluir células caliciformes y células M, que reaccionan inmunológicamente con patógenos y compuestos químicos.
Describa mejor la leyenda de la figura 1, incluyendo el nombre de las abreviaturas.
Both the figure and caption have been edited seeking improvement.
2) Describir con mayor detalle el mecanismo por el cual CYP1A permite mantener la integridad celular.
CYP1A, encontrado en mayúsculas y minúsculas, correcto.
Figura 2, la imagen es confusa, no se indica claramente la secreción de las células, el estímulo que reciben y que citoquinas les son propias. Es necesario utilizar flechas.
We edited and improved the figure 2 and standardized the CYP1A in the text.
More detail of the mechanism by which CYP1A allows to maintain cell integrity were introduced in the text
3) Es fundamental en un proceso inflamatorio explicar el efecto de los pesticidas u otras sustancias sobre la expresión del complejo inflamasoma, debe incluirse en la revisión.
The role of the inflammasome in the induction of inflammation by pesticides was included in the text.
4) Considero necesario hacer una descripción por familias de plaguicidas como los piretroides en cada ítem.
The description of pyrethroids has been added
5) es importante mencionar en la revisión el efecto de la disbiosis en el sistema nervioso.
The impact of pesticides on the digestive tract and function in the nervous system were included.

Round 2
Reviewer 1 Report
Thank you for correcting the manuscript!
Reviewer 2 Report
The study "Disruption of AhR signaling as a possible common link between pesticides, breakdown of epithelial barrier integrity, dysbiosis and chronic inflammatory diseases" has a significant improvement and the suggestions have been taken into account. I agree with the publication.